# Measurement Methods Used to Assess the School Food Environment: A Systematic Review

**DOI:** 10.3390/ijerph17051623

**Published:** 2020-03-03

**Authors:** Siobhan O’Halloran, Gabriel Eksteen, Mekdes Gebremariam, Laura Alston

**Affiliations:** 1Department of Nutrition, Institute of Basic Medical Sciences, University of Oslo, 0317 Oslo, Norway; mekdes.gebremariam@medisin.uio.no; 2Division Human Nutrition, University of Cape Town, UCT Medical Campus, Cape Town 7925, South Africa; gabriel.eksteen@uct.ac.za; 3Global Obesity Centre (GLOBE), Faculty of Health, Institute for Health Transformation, Deakin University, Geelong VIC 3217, Australia; laura.alston@deakin.edu.au; 4Deakin Rural Health, School of Medicine, Deakin University, Geelong VIC 3217, Australia

**Keywords:** school food environment, diet, measurement methods, INFORMAS, obesity, canteens, tuck shops, cafeterias

## Abstract

Children consume approximately half of their total daily amount of energy at school. Foods consumed are often energy-dense, nutrient-poor. The school food environment represents an effective setting to influence children’s food choices when dietary habits are established and continue to track into adulthood. The aim of this review was to: (1) group methods used for assessing the school food environment according to four food environment dimensions: Physical, economic, socio-cultural and policy and (2) assess the quality of the methods according to four criteria: Comprehensiveness, relevance, generalizability and feasibility. Three databases were searched, and studies were used to assess food and beverages provided at school canteens, tuck shops or cafeterias were included. The review identified 38 global studies (including 49 methods of measuring the food environment). The physical environment was the primary focus for 47% of articles, aspects of policy environment was assessed by 37% articles and a small number of studies assessed the economic (8%) and socio cultural (8%) environment. Three methods were rated ‘high’ quality and seven methods received ‘medium’ quality ratings. The review revealed there are no standardized methods used to measure the school food environment. Robust methods to monitor the school food environment across a range of diverse country contexts is required to provide an understanding of obesogenic school environments.

## 1. Introduction

The International Network for Food and Obesity non-communicable diseases Research, Monitoring and Action Support (INFORMAS) defines the food environment as the “collective physical, economic, sociocultural and policy surroundings and opportunities and conditions that influence people’s food and beverage choices and nutritional status” [1]. This broad frame-work definition is useful to identify the structural drivers of food acquisition, consumption and nutrient profiles and may encompass a number of measurable key dimensions (availability, accessibility, affordability and desirability of foods) to guide empirical research [2]. Food environment research has gained traction over recent decades in response to the role food environments have played in the global shift away from dietary patterns which included whole grains, fruits, vegetables and legumes to diets comprised of inexpensive, highly palatable and nutrient deplete ultra-processed foods [3]. A number of studies have shown the association between food environments and obesity, dietary patterns, chronic disease and other health related factors [4,5,6,7,8,9,10]. The World Health Organization (WHO) [11], the Institute of Medicine (IOM) [12] and the Centers for Disease Control and Prevention [13] have identified interventions to impact the food environment as strategies for creating population wide improvements in dietary patterns and weight status. The effect of food environments on dietary intake has also increasingly become a policy focus [14,15,16] set against the milieu of the Sustainable Development Goal 2 to end hunger, achieve food and nutrition security, improve nutrition and promote sustainable agriculture [17].

National survey data from high income countries show large numbers of children consume inadequate amounts of fruits and vegetables that fall well below recommended guidelines [18,19,20]. A recent Australian study, of 3496 school children showed that only 15% of males and 18.5% of female school children, reported consuming enough vegetables to meet the dietary guidelines [21]. Worldwide studies have also revealed that children consumed almost 40% of their daily recommended energy intake from energy-dense nutrient-poor foods (EDNP) [22,23]. 

The school food environment as defined by the United Nation’s Food and Agriculture Organization (FAO) “refers to all the spaces, infrastructure and conditions inside and around the school premises where food is available, obtained, purchased and/or consumed (for example tuck shops, kiosks, canteens, food vendors, vending machines” [24]. It represents an effective setting for interventions to influence children’s food choices at a time when dietary habits are developed and continue to follow an established trajectory across the lifespan [25,26]. Children consume on average, 40% of total amount of energy during the school hours [27] and often the types of foods consumed are high in saturated fat, added sugar and salt (e.g., waffles, chocolate milk, iced teas, cakes and sausages) [28]. The provision of these types of foods and beverages at these settings undermine dietary guidelines and encourage the notion that these types of foods may be consumed everyday rather than occasionally [29]. School canteens/tuck shops/cafeterias, which are often the sources of these types of foods, are also highly visible and accessible and thus play an important role in modelling a healthy food environment and establishing healthy eating behaviors early in life [22,27,30]. Furthermore, despite school canteen’s being an optimal intervention target, there is existing evidence that their adherence to government healthy nutrition policy is not always ideal [31].

The school food environment varies internationally with countries including Australia, Canada, New Zealand and South Africa providing food and beverages for purchase via canteens or tuck shops and others including the United Kingdom (UK) and the United States providing meals via school lunch programs [22,32]. This heterogeneity and complexity of the school food environment presents a challenge in determining a best practice, standardized measurement tool to assess the school food environment. 

High quality measurement methods are required to conduct assessments and evaluations of the school food environment to inform future policies and practices that will lead to improved quality of life, improvements in healthy weight and associated direct and indirect health care costs. Categorization of the methods used to measure the school food environment according to the conceptualized four dimensions of the food environment (i) physical (availability); (ii) economic (cost); (iii) socio-cultural (attitudes, perceptions) and (iv) policy (the rules) [33] may inform the choice and harmonization of the assessment methods and facilitate a greater understanding of obesity promoting school environments. 

In this context and to further the work in this area we undertook a study to: (1) classify the measurement methods according to the four dimensions of the food environment and (2) review and assess the quality of the measurement methods used to assess the school food environment 

## 2. Methods

This review followed the systematic steps outlined in the PRISMA guidelines [34] and was registered with the International Prospective Register of Systematic Reviews (PROSPERO) (registration number CRD42019125063).

### 2.1. Search Methods to Identify Studies

We conducted an electronic search in January 2019 of peer reviewed literature using: Medline, Embase and Web of Science. Recent cross-sectional studies conducted globally formed the basis for this review. Search terms were adapted to databases according to three main topic areas of (1) school food environments; (2) measurement methods and (3) tuck shop or canteen, café or cafeteria. For this review, the school could either be a primary/elementary school, secondary/senior school or after school care. 

### 2.2. Inclusion and Exclusion Criteria

The following search criteria were used in this review: (i) studies published in English in peer reviewed studies, (ii) human studies, (iii) studies from any country and (iv) studies which specified the methods or tools used to assess food and/or beverage items provided at school canteens, tuck shops or cafeterias. Studies included in this review included both qualitative and quantitative methods. 

Reviews, conducted in settings other than schools (e.g., kindergartens), focused on individual dietary intakes or nutritional status of children, assessed foods provided from vending machines or food wastage or primarily utilized dietary assessment methods (e.g., 24-h recall or food frequency questionnaires) were excluded.

### 2.3. Study Selection, Data Extraction and Analysis

The lead researcher (SO’H) reviewed all results from all three databases, removed duplicates and screened all results based on titles and abstracts against the review criteria (Figure 1). Reference lists of included articles were searched for additional relevant studies. When the abstract was considered insufficient to make conclusions about inclusion the full text was screened. 

Data were extracted into a detailed data spreadsheet from the full text independently by the two researchers (SO’H and GE). Any discrepancies were resolved by consensus with a third reviewer if necessary. 

### 2.4. Quality Assessment of Methods 

This study used an adapted version of a quality assessment tool that included criteria which was developed by INFORMAS researchers from public health and political science literature [1]. To provide an overall assessment of the quality of methods, the following four criteria were considered most relevant to critically assess the quality of the methods [35] used for measuring food provision in this context: Comprehensiveness, relevance, generalizability and feasibility. 

Methods were assessed against these criteria and the results combined to form an overall quality rating for each method (refer to Appendix A for more details of criteria and standards for quality assessment of methods used). Two independent reviewers (SO’H and MK) in a two-step process completed the quality assessment: The first reviewer assessed the quality of all studies; the second reviewer assessed the quality of a 10% random sample of the reviewed studies. A study sample size of 10% has been used previously for random sampling in a number of other studies [35]. The two reviewers were in consensus on the quality of all papers in the 10% sample. 

### 2.5. Dimensions of the Food Environment Framework 

In this study, we have used the four environmental dimensions (physical, economic, sociocultural and policy) as defined by Swinburn et al. [33] to categorize the methods used to measure the school food environment e.g., semi-structured interviews with food service providers regarding school nutrition guidelines was categorized as ‘policy’ food environment. (Table 1). 

## 3. Results

The extensive search yielded 18,350 relevant abstracts, with a total of 38 articles meeting the inclusion criteria (Figure 1). Major reasons for exclusion at full text stage was that the study did not explicitly describe the measurement method used to measure the food environment. An overview of key study characteristics is provided (Table 2). Articles were published from 2003 to 2018 and included data collected between 1993 and 2016. 

### 3.1. Where has Food Environment Research been Undertaken?

Of the 38 studies included, one study featured multiple countries including Northern/Central, Southern and Eastern-European countries [36] and 37 studies were single-country [22,28,37,38,39,40,41,42,43,44,45,46,47,48,49,50,51,52,53,54,55,56,57,58,59,60,61,62,63,64,65,66,67,68,69,70,71]. Thirty-three studies were conducted in high-income countries (87%) and five located in low and middle-income countries (13%) (Figure 2). 

### 3.2. Quantitative Methods 

Thirty-one articles sought to characterize the school food environments using quantitative methods. Amongst these articles the vast majority featured one measurement method including menu analysis (*n* = 5), direct observation (*n* = 3), self-completed questionnaire (*n* = 11) and surveys (*n* = 8). Four articles (15%) featured two approaches including surveys and canteen menu audits [65] or observations [60] menu analysis and telephone interviews [22] and self-reported survey and menu audits [66]. 

### 3.3. Qualitative Methods 

Of the 38 articles, two articles used qualitative stakeholder-based methods to describe the school food environment [46,57]. Drummond and Sheppard utilized focus groups interviews with Australian school students and semi-structured interviews with principals, parents, teachers and canteen managers. The interviews explored participant’s perceptions around healthy eating, likes/dislikes about the canteen, policy implementation and canteen profit expectations were probed [46]. The article by Masse et al., featured a single method, where Canadian principals and teacher/school informants participated in semi-structured interviews exploring their perceptions of the implementation of national guidelines within the school environment [57].

### 3.4. Mixed Method and Mixture of Methods

Five articles included a mixed methods approach where quantitative data was integrated with qualitative data, to provide comprehensive insight into the school food environment [28,37,39,40,42] and one study conducted a mixture of methods study [64]. Bevans and colleagues [39] and Ardzejewska et al. [37] used semi-structured interviews with food service managers/school principals/deputies where their school’s nutrition service policies and practices were explored, together with either student questionnaires [40] or canteen menu audits [37]. 

Two articles which featured focus groups, explored concepts around healthy eating with students from South Africa [39], and in the UK student discussions centered on four key themes: Children’s food environment, food intake, obtaining food and social aspects of food consumption [42]. South African learners also completed a self-administered questionnaire which examined tuck shop purchasing behavior whereas the school food environment was assessed by a lunch time observation and menu analysis in the UK [42].

Chortatos et al. [28] conducted student focus group discussions which explored themes such as eating habits, definitions of healthy and unhealthy foods and attitudes towards diet. Interviews around food availability and meals served at school were conducted with school administrators and a student web-based questionnaire included questions about school canteens and food and drink consumption [28]. 

Pettigrew et al. [64] conducted a mixture of methods study which included semi-structured focus groups discussions with parents around school food policy and school-based stakeholder (principals, teachers and canteen managers) interviews, which included knowledge and attitudes around canteen policy and factors influencing compliance with the policy. In the quantitative phase of the study, the parents and principals responded to a telephone questionnaire [64]. 

### 3.5. The Food Environment Dimensions, Quality Assessment and Advantages and Disadvantages

Table 2 provides study details including key features, advantages and disadvantages of the methods used to assess the school food environment, the food environment dimensions and the overall quality rating of the method. Only three studies received a ‘high’ quality rating [37,43,53]. These were studies conducted in high income countries (e.g., Norway and New Zealand) and included all four dimensions of the school food environment. Seven studies were rated as ‘medium’ quality [22,28,38,46,60,65,71] and twenty-nine studies were rated as ‘low’ quality [37,39,40,41,42,44,45,47,48,49,50,51,52,54,55,56,57,58,59,61,62,63,64,66,67,68,69,70]. The number of methods which were assessed was 49, as a number of studies applied more than one methodological approach to measure to school food environment.

#### 3.5.1. Physical Environment

The physical food environment was the primary focus for eighteen articles (47%) included in this review [38,39,42,45,47,48,49,50,52,53,54,59,62,63,67,68,69]. It was operationalized in terms of either availability of food items frequently for sale or the physical presence of a school food outlet. Three studies conducted onsite inspections of school canteens to determine either the types of ‘competitive’ foods (e.g., foods high in salt, sugar and saturated fat) sold [67] or the frequency of food and drinks from different food groups sold according to days/week [63] or the availability of foods items in à la carte restaurants [52]. Student surveys were used to determine the frequency of use of à la carte cafeterias [62] and tuck shops [59] and a self-administered tuck shop checklist was completed by teachers in a study by Ma and Wong [56]. Two studies determined the presence of a canteen/booth/school store either via a school principal questionnaire [53] or by onsite observation by research staff [50]. In the study by Gebremariam et al. students were asked about the frequency of daily canteen purchases [54]. Finch and colleagues surveyed schools to determine frequency of canteen purchases [48] and Faber et al. determined what food items were available from the school tuck shop in poorly resourced South African schools, via questionnaires and observations [47]. 

#### 3.5.2. Economic Environment

A small number of studies (*n* = 3) measured the economic environment which involved obtaining the price of foods available in school canteens either via extracting price data from online canteen menus (items available for purchase were snacks e.g., crisps, fruit and lunch e.g., sandwich, pizza snack) [41,71] or by obtaining canteen sales data from food service staff at participating schools [43]. Two studies utilized the data from the online menus, to conduct price analysis to determine if there were significant differences between the mean price of healthy and unhealthy products [41,71]. Carter and Swinburn described the cost of healthy and unhealthy foods sold in schools and determined if the canteen was run for profit, not-for-profit or contracted it out as a private business [41]. The authors also determined if schools used food sales for fundraising [43]. All three studies demonstrated that the pricing of foods sold in school canteens favored less nutritious foods compared to nutritious alternatives which provided some insight into the school economic food environment [41,43,71].

#### 3.5.3. Socio-Cultural Environment

Three studies measured aspects related to the school’s socio-cultural environment [41,47,53]. As part of a principal questionnaire, Gebremariam et al. assessed the perceived responsibility of the school for the diet of Norwegian students and the degree of priority given to food and nutrition beyond what was mandatory via a statement with a five-response category [53]. Faber and colleagues asked South African educators via a self-administered questionnaire about their interest and training received in nutrition, if nutrition was included in classroom teaching and their perceived role in nutrition education and healthy eating promotion [47]. In New Zealand, Carter and Swinburn developed a questionnaire using information from semi-structured interviews with primary schools and asked schools to rate three statements to indicate how they applied to their school: If nutrition was a priority, if healthy food provision was supported by management and if foods provided at school were highly nutritious [43]. 

#### 3.5.4. Policy Environment

The literature search yielded fourteen (37%) studies, which assessed aspects of the policy environment [70]. Most studies investigated if the school had a food policy either in combination with assessing the physical environment (e.g., the availability of food and drink) the socio-cultural environment (responsibility of the schools for children’s diets) or the economic environment. Only one study solely assessed the policy environment via principal interviews [37]. Twelve studies (29%) approached the school principal about their school food policy, one study questioned teachers [43], two canteen managers [65,66] and one examined online school menus to determine if food items available for purchase in canteens were compliant with guidelines [70]. Three studies determined if the school food policy was compliant with a government-mandated policy [28,65,70].

## 4. Discussion

This is the first review to comprehensively report on the methods used to measure the school food environment. The review identified 38 relevant studies which we categorized according to the conceptualized four dimensions of the food environment. The quality of the methods varied widely with only three methods [36,43,53] rated as high quality according to the detailed assessment criteria (Table 2) and included all four dimensions of the food environment. Studies that included food environment assessment methods that rated as high quality included a school management questionnaire of the food environment (guided by the existing Analysis Grid for Environments Linked to Obesity (ANGELO) framework [33]), school environment assessments and the collection of food sales data from school food outlets.

The most common method used to measure the school food environment was a self-administered questionnaire/survey (*n* = 21) [28,36,39,40,42,43,44,45,47,48,51,52,53,54,56,58,59,62,64,66,68] to obtain information such as food policy, food purchased or availability from tuck shops or school meals, presence of a tuck shop, if nutrition training was a priority or attitudes to nutrition from either school principals, students, educators or canteen managers. Disadvantages with this method include self-reported bias in favor of desirable rather than actual practice, a low response rate which may not be indicative of true food provision and respondent burden. Likewise, survey results may not capture realistic practices e.g., food policy as an indicator of the policy environment does not assess the level of effectiveness of a policy [43]. Advantages are the low cost, ease of administration, access to a large number of participants and the reporting of other nutrition practices that may be otherwise overlooked by just reviewing menus. 

Six of our included studies utilized observational data collection methods where trained researchers observed food provision in the canteens/tuck shops or school restaurants [37,38,47,60,65,67]. This data collection method can be highly variable and can subsequently provide inaccurate reflections of food provision and poor generalizability of findings [72]. However, direct observation methods may improve the validity of self-administered surveys and on-site/direct observations are considered the ideal approach in assessing school environmental characteristics [60] and nutrition practices [32].

Canteen menu analysis was utilized by six studies, which involved either obtaining a school menu online or from canteen managers [22,55,61,65,70,71]. Although menu review is an objective measure, it may not be a reliable tool for food provision assessment, as the actual food available may differ from the planned menu and insufficient information may limit the account of portion sizes, types of foods, or pricing [64]. Skilled researchers (e.g., dietitians) in menu coding and analysis are also often required to conduct the research and often menus are only assessed at one point in time, so the certainty of the menus being offered at all times, over the school year is unknown. However, compared to on-site observations, menu reviews are lower in implementation costs and less labor-intensive [64]

Seven studies included semi-structured interviews with school principals or canteen/food service manager [41,46,50,57,60,64,69] and/or focus groups with students [46,64]. This subjective measurement method is expensive due to the labor required to conduct the research. However, rich data can be obtained, and other food provision practices may be captured that may be otherwise missed by observational methods. 

Of note, only small number of studies (*n* = 7) included validity and/or reliability tests for the methods used to measure the school food environment, the details of which are included in Table 2. 

Of the four conceptualized dimensions, the physical food environment was the most common dimension investigated. A medium to low quality assessment rating was applied to most studies due to utilization of only one measurement method (e.g., the availability of food items for sale). Inclusion of more than method to measure the physical food environment (e.g., indication of the physical presence of a purpose built canteen, the availability of food for sale and the frequency of sale) would have gained an optimal quality rating. A shared theme across most studies was the availability of cheap, convenient foods high in saturated fat, salt and sugar available at school canteens/tuck shops. There is considerable scope to improve the availability of healthy choices, within schools in view of reducing the risk of childhood overweight and obesity. Indeed, limiting or decreasing the availability of unhealthy foods has been found to be associated with less frequent purchases of these items in school [62]. Other study findings relate to the physical presence of school canteens with one study noting that the presence of school canteens did not seem to influence children’s intake, although the authors did note that these findings may be attributable to the infrequency in canteen opening times (most were only open once/week) [53]. Another reported that whilst most schools had a canteen food service, many had inadequate canteen facilities, with only 15% of schools having purpose-built facilities [43]. The lack of canteen amenities limits the capacity to provide freshly prepared foods and encourages the sale of prepackaged products such as pies and other savory pastries [43]. 

Regarding the economic environment, one study found that schools operated their food service for profit and used food products (pizzas and pies) as fundraising initiatives [43], suggesting that school profitability was placed above student health as a priority. It is also worth noting that only three studies investigated the cost of healthy versus unhealthy foods and showed that the mean price of healthy lunch items (e.g., salad sandwich) was greater than the less healthy item (e.g., meat pie) [41,43,71] which suggests there is an opportunity to introduce pricing strategies to make healthier choices the easiest options for children. Price is seldom considered in healthy school food policies, yet studies have shown that improving youth’s financial access to healthy foods may reduce the risk of obesity, particularly amongst those from a lower socio-economic position [73]. Price is also a strong predictor of consumer’s choice of food and beverages and given that schools have a duty to create a healthy school food environment [74] further investigation of pricing strategies of school canteens/tuck shops could be a focus for future research, particularly in lower-income schools which are reported as having less healthy food environments than in high-come schools [75]. In terms of limitations of the findings, all three included studies utilized descriptive analysis. Thus, the examination of the associations between student purchases and the price of school canteen food would provide a more comprehensive picture of the economic food environment. Inclusion of price changes over time, the effect of seasonal discounts and interactions between price and promotion/or placement of canteen foods would improve the quality of the methods used to measure the economic food environment. 

Only three studies included in our review solely investigated the socio-cultural environment [43,47,53]. Based on the quality assessment, measuring the socio-cultural environment via questionnaires regarding attitudes to nutrition was limited by respondent bias. Collectively, schools had positive attitudes to nutrition, acknowledged a responsibility for children’s diets and agreed that health and nutrition should be prioritized and promoted by the school. It is unclear whether these findings translate into successfully influencing children’s healthy eating knowledge and improved eating habits and thus warrants further research. Of note, one study reported that schools did not consider the school environment as playing a part in nutritional outcomes which may be because educators believe they have limited influence on altering the food service [43]. It is important that the attitudes and beliefs of school food providers, which can be seen as potential barriers to the implementation of school food policies [66], should not be overlooked when analyzing the environmental factors influencing obesity and ought to be used to guide implementation support strategies [34]. 

The policy environment was a dimension that was explored by a number of articles included in our review which probed the existence of a school food policy governing food availability or if the foods choices available in school canteens were compliant with government mandated policies and guidelines. A common finding across our studies revealed that nutrition policies are poorly implemented at the school level, limiting their public health impact [22,55,70]. 

The WHO Global Strategy on Diet, Physical Activity and Health has called for “governments to adopt policies that support healthy diets at schools and to limit the availability of products high in salt, sugar and fats” [76]. In addition, the development of nutrition standards for all foods sold or provided at schools has been recommended by a number of agencies including the WHO [77], the Center for Disease Control and Prevention [13], the IOM [12] and the WHO EU [77]. In keeping with these recommendations, the WHO developed the Nutrition Friendly Schools Initiative, where schools which meet a set of criteria will be recognized as “Nutrition Friendly Schools”. Although the initiative does not provide specific standards related to the nutritional quality of foods provided and sold, it is a whole-of-school approach that calls for healthy diet and eating practices [78]. Policies have also been introduced in the school setting in a number of countries that support the provision of food aligned with national dietary guidelines. For example, all Australian states and territories have introduced voluntary healthy canteen policies that promote the sale of healthy foods and restrict the sale of less healthy foods [70]. Likewise, in the UK the mandated ‘School Food Plan’ is a set of standards that compels schools to provide children access to nutritious meals at schools [79]. However, international research suggests that most schools fail to implement school nutrition policies [80,81] and on their own, they are insufficient to guarantee the provision of healthy foods in schools. To ensure policies are appropriately implemented to achieve the desired outcomes, to contribute to accountability measures to stakeholders and to provide a basis for future actions, comprehensive monitoring of adherence to school food policy standards is required. 

### 4.1. Future Practice

The International Network for Food and Obesity non-communicable diseases Research, Monitoring and Action Support (INFORMAS) is a global network of researchers that aims to monitor benchmark and support public and private sector actions to create healthy food environments and reduce obesity and non-communicable diseases (NCDs) [1]. The network aims to do this by developing a global framework for monitoring foods and beverages provided or sold in schools that can be used to compare and evaluate the nutritional quality of the foods, compared with specific policies within and across jurisdictions. The framework is composed of nine impact modules one of which ‘food provision (in schools)’, where information about nutrition policies in countries and school nutrition policies and standards/guidelines are collected in two components. These monitoring practices facilitate accountability measures and provide a basis for the development of new or improved standards. In addition, nutrition policy programs are also important for other areas of research such as local food environments surrounding schools [82,83] and associated environmental impacts and economic development [84].

Assessing adherence to nutrition standards in some countries will still be a challenge as policies/guidelines differ in the way they have been developed (voluntarily or mandatory) and implemented at different government levels (e.g., national, state/provincial or local) [32]. Guidelines may differ in the way they are applied e.g., just to meals/foods served or available for purchase or the whole school food environment to include fundraising and sponsorship [33]. Monitoring therefore may also be difficult in some countries or states where food is not centrally provided by schools or in low to middle income countries which may not have the financial capacity to monitor nutrition programs. Even so, a major advantage of applying the INFORMAS framework is the use of a standardized method to measure the school food environment in different contexts which will allow for comparisons across different countries, settings and times. There is also the potential for effective benchmarking of performance, which can assist in contributing to increasing accountability of schools and their actions to improve the healthiness of the foods provided. Some progress within the ‘school provision’ module has been made, with an online tool called the School Food Environment Review and Support Tool (School-Ferst), available on the INFORMAS site, which is designed to support schools to assess the healthiness of the foods and beverages provided (available online: https://www.informas.org/modules/food-provision/). 

Another measurement tool not identified in the studies included in this review, which could be applied to the school food environment, is the Food Store Environment Examination (FoodSee) methodology, which quantifies participant’s interaction with the food store environment [85]. Participants wear a camera and global positioning system (GPS) unit on a lanyard, which captures 136-degree image of the scene ahead approximately every seven seconds, enabling accurate and rapid speed mapping of the surrounding food environment in the participant’s location [85]. This new tool has been utilized in a feasibility study, which focused on images from food outlets captured by children aged 11–13 years and evaluated the possibility of assessing food availability and marketing [85]. This method requires further validation but shows promise of being a high-quality methodology to measure surrounding food environments, particularly the availability of food. For studies involving children, this tool could be utilized to objectively and unobtrusively measure children’s interaction with school canteens/tuck shops and could enable a number of the four dimensions of the food environment to be assessed e.g., food product availability, placement, price promotion and purchases.

In addition, it important to note, that our review revealed only 38 global studies which assessed the school food environment with dominance in high income countries (e.g., USA, Australia). Thus, if the World Health Organization, INFORMAS and/or government organizations are to make substantial gains in improving the school food environment globally, then there is a need for further research into appropriate measurement methods which are high quality and can be applied broadly across a range of country contexts. 

### 4.2. Strengths and Limitations 

There are a number of strengths to this review. A comprehensive search strategy of literature, using robust review methods was undertaken to identify methods used to measure the school food environment. The study rated the quality of each method and the quality assessment criteria could be applied elsewhere. However, this study has several limitations. Firstly, the search was restricted to English language publications, which may have resulted in the exclusion of important non-English publications. Studies utilizing methods to measure the school food environment were mostly from high-income countries rather than low and middle income countries. This may have been due to the literature search being limited to peer reviewed studies in English only and as such relevant publications in languages other than English may have been missed. In addition, studies were conducted in different contexts making comparisons challenging. 

## 5. Conclusions

Grouping the measurements methods according to the four dimensions of the school food environment provided insight into which dimensions were most commonly explored and those elements that may warrant further research in the future. Our review also revealed that there are no common standard methods used to measure the school food environment across different country contexts. This was due to the diverse methodological approaches used to measure the school food environment and the differing jurisdictions where foods are provided to children as part of a school lunch program (e.g., the UK) or where foods are available for purchase at school canteens or tuck shops (e.g., Australia). The lack of standardized measurement methods identified in this review is broadly consistent with previous systematic reviews from high income [4,10,86,87,88] and low–middle income countries [89] and with a review which quantified the methods used to measure the ‘retail food environment’ and associations with obesity [90]. The field therefore is in need of standardized methods and indicators to profile and monitor the school food environment across the diverse high and low to middle income settings and to provide robust assessments of the influence of the school food environment on nutrition and health. 

## Figures and Tables

**Figure 1 ijerph-17-01623-f001:**
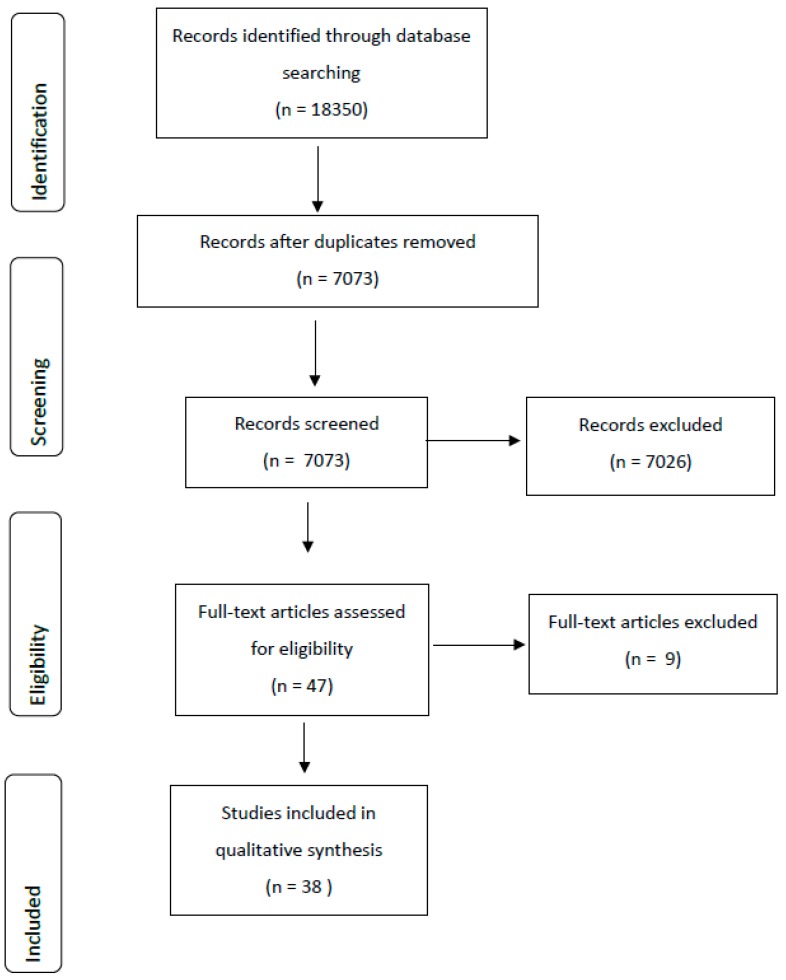
Summary of the literature search process.

**Figure 2 ijerph-17-01623-f002:**
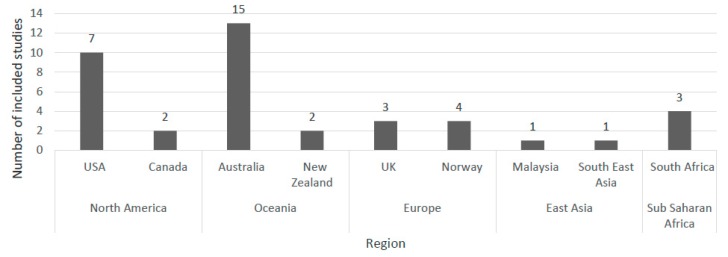
Study characteristics of the articles identified in the review.

**Table 1 ijerph-17-01623-t001:** The four environmental dimensions used to group the food environment measurement methods (adapted from Swinburn et al. [33]).

Dimensions	Description
The physical environment (what was available?)	Purpose built canteens, inadequate facilities, no facilities. Food items available in the canteen recorded, frequency of sale
The economic environment (what were the financial factors?)	Food service ran for profit, contracted out to a private business or ran for not-for-profit. Price of healthy and unhealthy food items recorded, children’s spending restriction, product placement and price of items sold in the canteen.
The socio-cultural environment (what were the attitudes and perceptions?)	Who are the people responsible for canteen—nutrition knowledge/training. Is nutrition high on the list of priorities? Management is supportive of healthy food provision Food service provides mainly foods with high nutritional value
The policy environment (what were the rules?)	Does the school have a food policy that specified the types of food and drinks and promotion and pricing of products in canteens? If so, how affective is that school policy? Who regulates this school food policy?

**Table 2 ijerph-17-01623-t002:** Summary of identified studies measuring the school food environment.

Author	Study Country	Objective of the Study	Setting	Dimension of Food Environment Measured	Description of Methods (Menu Analysis, Questionnaires)	Limitations/Strengths of Design and Methods Food Environment Measures Used	Overall Quality of Method Used to Measure Food Environment
Ardzejewska et al., 2012 [37]	Australia	To investigate the barriers and facilitators to, and the extent of the implementation of, the New South Wales (Australia) ‘Healthy School Canteen Strategy’	Two primary and two secondary government schools from a low socio-economic region in Sydney, New South Wales (NSW), Australia	Physical and policy	The existence of implementation of the New South Wales ‘Healthy School Canteen Strategy’ was measured through both a quantitative audit and qualitative interviews	Quantitative audit (observational cross-sectional survey—self report bias) Qualitative interviews face-to-face semi-structured interviews provided rich data, and context specific Validity and reliability not tested	L
Beets at al., 2016 [38]	USA	To assess compliance with internal healthy eating standards	YMCA-operated After school program children aged 5–12 years	Policy, economic and sociocultural	Direct observation of food provision and staff behavior	Direct observation from researchers a strength of the study along with survey methods	M
Bekker et al., 2017 [39]	South Africa	To investigate tuck shop buying behavior, choices of lunchbox items and healthy eating perceptions and attitudes at two schools	Grade 2 to 7 students from a school with a nutritionally regulated tuck shop and a school with a conventional tuck shop. Bloemfontein, South Africa	Physical, economic	Self-administered questionnaire on money spent, foods purchased, lunchboxes, likes, attitudes Focus group discussions based on discussion guide on perceptions, attitudes and behaviors	Self-reported bias with student’s questionnaire. Content validity of questionnaire tested with teachers and dietitians. Face validity determined by study researcher. Reliability of questionnaire tested by a pilot study. Cronbach’s α value +0.92	L
Bevans et al., 2010 [40]	USA	To test if the availability of nutritious foods during school lunch periods, as indicated by compliance with USDA recommendations, would be positively associated with children’s healthier eating behavior both in and out of school.	22 schools 2039 participants grades 5–8	Physical, policy, economic	Students answered a self-administered questionnaire about perceptions, attitudes, buying behaviors and lunchbox content	Self-report bias, accuracy of children’s reporting in questionnaires. Validity and reliability not tested	L
Billich et el., 2018 [41]	Australia	To examine the relative price of ‘healthy’ and ‘less healthy’ lunch and snack items available within school canteens	200 primary and secondary government schools	Economic	Observational cross-sectional study using semi structured interviews with food service managers and schools and students (student reporting of nutrition program participation and eating behaviors)	Semi-structured interviews provided rich data. Menu analysis objective measure. Only menu items that could be conclusively classified as ‘green’ (foods recommended to eat most) were included in the price analysis. Only canteen menus available online were included in the analysis. The cheapest menu item from lunch and snack categories used to represent the most affordable comparison. Validity and reliability not tested	L
Briggs et al., 2011 [42]	UK	To analyze and describe 8–10-year-olds’ home and school food environments	*n* = 24 children *n* = 18 parents 8–10 years old 1 school lower SES area	Physical	1. Photographic food diary for 4 days 2. focus group discussions based on photos (used for quantitative and qualitative analysis) 3. Home Food Environment Questionnaire (HFEQ) for self-completion by parents. (addressed consumption patterns and eating behaviors 4. informal observation of school’s dining room 5. analysis of menu and discussion with cook for conformance with school food trust’s food based standards	Mixed-method approach of qualitative and quantitative data was used. Observation by survey method. Qualitative focus groups of dietary intake provided detail intake information and reduced self-report errors in standard survey. Validity and reliability not tested	L
Carter and Swimburn, 2004 [43]	New Zealand	To identify and quantify the potential impact of environmental factors on the promotion of unhealthy weight gain.	National primary schools	Physical, policy, economic, socio cultural	School environment questionnaire - to assess key elements of the physical, economic, socio cultural, policy environments for schools. School food sales also included as index of food eaten	Respondent burden via obtaining food sales data from food service staff. Measuring the difference in price between the ‘more’ and ‘less’ healthy foods provided a robust indicator. Presence or absence of a food policy important but assessing the effectiveness of the policy was subjective. Respondent bias via questions regarding attitudes to nutrition. Face validity of the questionnaire was tested with school teachers	H
Chortatos et al., 2018 [28]	Norway	To gain a better understanding of the consumption habits of adolescents in the Norwegian school lunch arena	12 secondary schools	Socio cultural	Qualitative focus groups with students. School staff were interviewed about adherence to guidelines for school meals Survey method: online questionnaires	Mixed methods provided a more comprehensive picture of the food environment. Self-reported data prone to respondent bias. Validity and reliability not tested	M
Cleland et al., 2004 [44]	Australia	To describe foods purchased from school canteens, and perceptions about school canteens from students, parents and teachers	12 primary schools	Socio cultural	Survey was used to obtain information from students, parents and teachers through self-completion questionnaires	Self-reported data prone to respondent bias. Validity and reliability not tested	L
Condon et al., 2009 [45]	USA	To describe foods offered in school meals and consumed by children	Samples included 130 school food authorities, 398 schools, and 2314 children (grades 1–12)	Physical	School Nutrition Dietary Assessment Study. School menu surveys were used to identify the foods offered in school meals	Menu analysis objective measure was a strength of the study. Validity and reliability not tested	L
Drummond and Sheppard, 2011 [46]	Australia	To investigate school canteens and their place within the school system	14 schools School principals *n* = 14, canteen managers *n* = 14, parents *n* = 50, teachers *n* = 10 and students *n* = 450 (aged 5-16 years	Physical	Qualitative study—interview, focus groups	Qualitative interviews and focus groups provided rich and context specific data Validity and reliability not tested.	M
Faber et al., 2013 [47]	South Africa	To assess the school food environment in terms of breakfast consumption, school meals, learners’ lunch box, school vending and classroom activities related to nutrition	90 poorly resourced schools	Policy, physical	Questionnaires were completed by school principals (*n* = 85), school feeding. The school menu (*n*= 75), meal served on the survey day, and foods at tuck shops and food vendors (*n* = 74) were recorded.	Direct observation a strength of the study along with survey methods. Questionnaires subject to reporting bias. Face validity of the questionnaires were tested with educators	L
Finch et al., 2006 [48]	Australia	To identify sources of food eaten during the school day, the types of foods and frequency of purchases from the canteen and association with SES and weight status in primary school-aged children.	18 government primary schools	Physical	Questionnaire with items relating to food and drink habits assessed usual canteen purchasing times and purchasing frequency, canteen spending, frequency of canteen use, sources of food and drink consumed at school and at breakfast, and types of food and drinks purchased	Self-report bias of questionnaire. Other important measures e.g., price differential of (‘healthy’ and ‘less healthy’) not collected. Questionnaire development included content analysis, pilot testing and reliability testing	L
Finch et al., 2007 [49]	Australia	To identify sources of food eaten during the school day, the types of foods and frequency of purchases from the canteen and association with SES and weight status in primary school-aged children.	16 Primary schools (8 high SES, 8 low SES), 2224 students average age 9.6 y	Physical	Students in years 4-6 completed a self-administered questionnaire (School Eating Habits and Lifestyle Survey), with parents completing the questionnaire on behalf of children in years 1-	Self-report bias, accuracy of student’s reporting in questionnaires. Survey test for reliability with a mean kappa 0.529 using pairings from 17 questions	L
Fox et al., 2008 [50]	USA	To examine the association between school food environments and practices and children’s body mass index (BMI; calculated as kg/m2).	The study included 287 schools and 2,228 children in grades 1 through 12.	Policy, physical	Data on school food environments and practices were collected through on-site observations and interviews with school principal	Direct observation and principal interview was a strength of the study. Validity and reliability not tested	L
French et al., 2003 [51]	USA	To describe food related policies and practices in Secondary schools in Minnesota	336 schools 463 school principal	Policy, physical	Survey questionnaires around school food environment and food related practices	Self-reported bias was a limitation. Validity and reliability not tested	L
French et al., 2003 [52]	USA	To describe the food environment in 20 Minnesota secondary schools.	20 schools	Policy, physical	Surveys mailed to school principals and food service directors	Self-reported bias was a limitation. Validity and reliability not tested	L
Gebremariam et al., 2012 [53]	Norway	To investigate the influence of the school food environment on the dietary behaviors of 11-year-old Norwegian children in elementary schools.	1425 11-year-old children from 35 schools	Policy, physical and socio cultural	School principal questionnaire modified from a nationwide school survey, covered different aspects of the school food environment	Self-reported bias was a limitation. Validity and reliability not tested	H
Gebremariam et al., 2016 [54]	Norway	To explore individual, home, and school/neighborhood environmental correlates of dietary behaviors (intake of fruits, vegetables, soft drinks, and unhealthy snacks) among adolescents.	742 adolescents with a mean age of 13.6. 11 secondary schools	Policy, physical and socio cultural	A web-based questionnaire for students. School teachers administered the questions to the students	Self-reported bias. Reliability tested for questions relating to school food/nutrition. Internal consistency scale 0.78	L
Hills et al., 2015 [55]	Australia	To describe the changes in school canteen food between 2007 and 2010 and characterizes schools most likely to adhere to strategy guidelines.	In 2007 265 schools provided menus; in 2010 95 schools provided a menu	Policy, physical	Menu analysis and canteen managers were asked for recipe information, product size and brand. The study examined changes over time in adherence to a healthy canteen policy;	Limitation: study did not capture sales data. Menu analysis objective measure. Validity and reliability not tested	L
Lien et al., 2014 [36]	Multi European	To describe practice within physical, political and sociocultural aspects of the school nutrition environment in seven countries across Europe based on questionnaires to the school management, and exploring their associations with soft drink consumption reported on questionnaires by 10–12 year olds.	A total of 160 schools responded to the school management q and 171 audits were conducted	Policy, physical and socio cultural	School Management Questionnaire (SMQ)—developed to assess the four types of the school environment according to the ANGELO framework	Self-reported bias of questionnaire. Validity and reliability not tested	H
Ma and Wong, 2017 [56]	Hong Kong	An examination of the relationship between available food in secondary school tuck shops and students’ purchasing preferences.	6 secondary schools, 374 students	Physical	Questionnaire which was adapted from the Department of Health Hong Kong. Food was categorized into prepared snacks, fresh and cooked foods and drinks. The checklist included recording items sold in tuck shops	Self-reported bias of questionnaire. Validity and reliability not tested	L
Masse et al., 2013 [57]	Canada	To explore the factors that impeded or facilitated the implementation of publicly mandated school based PE and nutrition guidelines in the province of British Colombia.	50 schools (17 principals and 33 teachers)	Physical	Semi structured interviews with principals and teachers	Interviews provided rich and context specific data. Validity and reliability not tested	L
Masse et al., 2014 [58]	Canada	To examine associations between the school food environment, students’ dietary intake, and obesity in British Columbia (BC), Canada.	174 principal responses and 11,385 students (7–12 grades)	Physical	School principals completed the school environment survey	Interviews provided rich and context specific data. Validity and reliability not tested	L
Moore and Tapper, 2008 [59]	UK	To estimate the impact of school fruit tuck shops on children’s consumption of fruit and sweet and savory snacks.	43 primary schools children aged 9–11 years	Physical	Student surveys	Self-report bias of surveys. Validity and reliability not tested	L
Nathan et al., 2013 [60]	Australia	To assess the validity of Principal self-report of primary school healthy eating and physical activity environments.	Primary school Principals (*n* = 42)	Physical, economic	Principal telephone interview, teachers observed food available in canteens and used for fundraisers	Observational data over a 9 week period was a major strength. Canteen food questions tested for validity and had a Kappa/PABAK score of range −0.6–0.81	M
Nathan et al., 2016 [61]	Australia	To examine whether a theoretically designed, multi-strategy intervention was effective in increasing the implementation of a healthy canteen policy in Australian primary schools.	51 Primary school (children aged 5–12 years)	Policy and physical	Menu analysis	Menu analysis objective measure was a strength of the study. Validity and reliability note tested	L
Neumark-Sztainer et al., 2005 [62]	USA	To examine associations between high school students’ lunch patterns and vending machine purchases and the school food environment and policies.	A randomly selected sample of 1088 high school students from 20 schools completed surveys about their lunch practices and vending machine purchases	Policy, physical	Data on school food policies were collected with surveys that were mailed to principals and food service directors	Self-report errors or bias of survey. Validity and reliability not tested	L
Nicholas et al., 2013 [63]	UK	To assess lunchtime provision of food and drink in English secondary schools and the choices and consumption of food and drink by pupils having school lunches, and to compare provision in 2011 with that in 2004.	A random selection of 5969 pupils having school lunches in a nationally representative sample of eighty secondary schools in England.	Policy, physical	Onsite school inspections of canteens	Direct observation major strength of study. Validity and reliability not tested	L
Pettigrew et al., 2013 [64]	Australia	To identify school principals’ perceptions of factors that influenced schools’ compliance with the new school nutrition policy and factors related parents’ beliefs about whether their children’s diets were healthier as a result of the policy.	Ten government primary and secondary schools	Policy	Semi structured interviews; focus groups with parents, interviews with school principals, teachers and canteen managers. Parents and principals questionnaires	Semi-structured interviews provided rich data. Validity and reliability not tested	L
Reilly et al., 2016 [65]	Australia	The aim of this study is to assess the validity and direct cost of four methods to assess policy compliance: (1) principal and (2) canteen manager self-report via a computer-assisted telephone interview; and (3) comprehensive and (4) quick menu audits by dietitians, compared with observations.	50 Primary schools (5–12 y o)	Policy and physical	Principal and canteen manager self-report—CATI interview. Observations of canteen food and beverages. Comprehensive menu audit and quick menu audit	Self-reported measured only consisted of one item. Self-reported bias. Direct observations a major strength. comprehensive Validity of quick menu audit kappa 0.68 and menu audit kappa 0.42	M
Reilly et al., 2017 [66]	Australia	To assess a range of barriers, as reported by canteen managers, using a quantitative survey instrument developed based on a theoretical framework.	A survey of 184 primary school canteen managers	Policy	Survey items assessed canteen manager employment status, canteen characteristics and potential barriers to healthy canteen policy implementation	Survey prone to participant bias. Validity and reliability not tested	L
Rosmawati et al., 2017 [67]	Malaysia	To determine the types of competitive foods sold in primary school canteens for the consumption of school children in Kelantan, Malaysia.	16 randomly selected primary school canteens	Physical	Site visit inspection was carried out to observe the running of the food preparation up to display of cooked or ready-to-eat food.	Direct observation was a major strength of study. Validity and reliability not tested	L
Temple et al., 2005 [68]	South Africa	To determine the food consumption patterns of adolescent students at schools.	476 students mean age 14.5 y	Physical	Student questionnaire about types of foods purchased	Questionnaire prone to participant bias. Validity and reliability not tested	L
Utter et al., 2007 [69]	New Zealand	To describe the demographic characteristics and food choices of school canteen/tuck shop users.	3275 students aged 5 to 14	Physical	Student interviews and the FFQ were administered at students’ homes; parents helped to complete the interviews and FFQ	Semi-structured interviews provided rich data. Validity and reliability not tested	L
Woods et al., 2014 [70]	Australia	To assess the compliance of school canteens with their state or territory canteen guidelines.	263 school menus were sourced and assessed	Policy	Menu analysis. Menu items were coded into one of 3 categories based on the traffic light system set by each specific state’s canteen guidelines; ‘green’ or ‘amber/red’. When it was uncertain whether a food should be classified as ‘green’ or ‘amber’ it	Menu analysis limited to schools with menus available online. Menus assessed at one point in time so unclear if foods were available throughout the whole year. Menu analysis objective measure. Validity and reliability not tested	L
Wyse et al., 2017 [71]	Australia	To describe the price of Australian school canteen foods according to their nutritional value.	70 primary schools	Physical, economic	Menu analysis—Menu items were coded into one of 3 categories based on the traffic light system set by each specific state’s canteen guidelines; Price data was also extracted	Menu analysis objective measure was a major strength of the study. Validity and reliability not tested	M
Yoong et al., 2015 [22]	Australia	To examine the availability of healthy food and drinks, implementation of pricing and promotion strategies in Australian primary school canteens, and whether these varied by school characteristics	203 Canteen managers telephone interview & 170 menus	Physical, economic	School canteen managers from primary schools telephone interview and provided canteen menus	Direct observation was a major strength of the study. Validity and reliability not tested	M

Notes: Abbreviations: ANGELO; Analysis Grid for Elements Linked to Obesity, CATI; Computer Assisted Telephone Interview, FFQ; Food Frequency Questionnaire, HKG; Hong Kong, USA; United States of America, SMQ; Senior Management Questionnaire, USDA; United States Department of Agriculture.

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
