# Peer review of "Measurement Methods Used to Assess the School Food Environment: A Systematic Review"

_ijerph, 2020, doi:10.3390/ijerph17051623_

Round 1
Reviewer 1 Report
The current study has showed that: 1) there are no standardised methods used to measure the school food environment; 2) robust methods to monitor the school food environment across a range of diverse country contexts is required to provide an understanding of obesogenic school environments.
Authors are kindly requested to emphasize the current concepts about these issues in the context of recent knowledge and the available literature. This articles should be quoted in the References list.
References
A systematic review employing the GeoFERN framework to examine methods, reporting quality and associations between the retail food environment and obesity. Health Place. 2019 May;57:186-199. doi:10.1016/j.healthplace.2019.02.007. Food store environment examination - FoodSee: a new method to study the food store environment using wearable cameras. Glob Health Promot. 2019 Aug 26: 1757975919859575. doi: 10.1177/1757975919859575.Author Response
Measurement methods used to assess the school food environment: A Systematic Review
To the reviewers and editorial team,
Thank you for your comprehensive review of our manuscript. Thanks to your comments, we feel the manuscript is significantly improved. Please see below your comment, our response and a reference to the manuscript with tracked changes.
With many thanks,
The authors.
Reviewer 1
The current study has showed that: 1) there are no standardised methods used to measure the school food environment; 2) robust methods to monitor the school food environment across a range of diverse country contexts is required to provide an understanding of obesogenic school environments.
Comment 1: Authors are kindly requested to emphasize the current concepts about these issues in the context of recent knowledge and the available literature. This articles should be quoted in the References list.
References
A systematic review employing the GeoFERN framework to examine methods, reporting quality and associations between the retail food environment and obesity. Health Place. 2019 May;57:186-199. doi:10.1016/j.healthplace.2019.02.007.
Food store environment examination - FoodSee: a new method to study the food store environment using wearable cameras. Glob Health Promot. 2019 Aug 26: 1757975919859575. doi: 10.1177/1757975919859575.
Response 1:
Thank you for this suggestion, we agree with reviewer 1. We have now included the Wilkins et al. review.
Reference in the manuscript with tracked changes: Page 31, Lines 462-463:
The lack of standardized measurement methods identified in this review is broadly consistent with previous systematic reviews from high income [4, 10, 865-887] and low-middle income countries [89] and with a review which quantified the methods used to measure the ‘retail food environment’ and associations with obesity [90].
We have also included the paper by McKerchu et al. in our discussion section
Reference in the manuscript with tracked changes: Page 30, Lines 421-434:
Another measurement tool not identified in the studies included in this review, which could be applied to the school food environment, is the Food Store Environment Examination (FoodSee) methodology, which quantifies participant’s interaction with the food store environment [85]. Participants wear a camera and Global Positioning System (GPS) unit on a lanyard, which captures 136-degree image of the scene ahead approximately every seven seconds, enabling accurate and rapid speed mapping of the surrounding food environment in the participant’s location [85]. This new tool has been utilised in a feasibility study, which focused on images from food outlets captured by children aged 11-13 years and evaluated the possibility of assessing food availability and marketing [85]. This method requires further validation but shows promise of being a high quality methodology to measure surrounding food environments, particularly the availability of food. For studies involving children, this tool could be utilised to objectively and unobtrusively measure children’s interaction with school canteens/tuckshops and could enable a number of the four dimensions of the food environment to be assessed e.g. food product availability, placement, price promotion, purchases.
Reviewer 2 Report
This paper addresses a valuable tool that can be used to evaluate access to healthy foods in schools. The paper is well referenced and I applaud the rigorous process that was applied for the inclusion and exclusion criteria. Including a method for assessing whether nutrition was included in the school curriculum would have been a value-added tool as well.
Overall, this was a good paper appropriate for local authorities as well for those interested in improving school food environments in countries throughout the world. The research could be made even more robust by including some means of measuring whether topics on nutrition or nutritional assessments were included in the curriculum or available as an independent tool.
Author Response
Measurement methods used to assess the school food environment: A Systematic Review
To the reviewers and editorial team,
Thank you for your comprehensive review of our manuscript. Thanks to your comments, we feel the manuscript is significantly improved. Please see below your comment, our response and a reference to the manuscript with tracked changes.
With many thanks,
The authors.
Reviewer 2
This paper addresses a valuable tool that can be used to evaluate access to healthy foods in schools. The paper is well referenced and I applaud the rigorous process that was applied for the inclusion and exclusion criteria. Including a method for assessing whether nutrition was included in the school curriculum would have been a value-added tool as well.
Comment 1: Overall, this was a good paper appropriate for local authorities as well for those interested in improving school food environments in countries throughout the world. The research could be made even more robust by including some means of measuring whether topics on nutrition or nutritional assessments were included in the curriculum or available as an independent tool.
Response 1:
Reference in the manuscript with tracked changes: page 2, lines 53-36.
Thank you for this suggestion. We agree with the reviewer that a research focus on measuring whether topics on nutrition or nutritional assessments were included in the curriculum or available as an independent tool. We feel this warrants a separate systematic review, with a specific focus on school curriculum and nutrition intake assessment in schools. In this review, our aim was to understand the methods used to measure school food environments (as defined by the United Nation’s Food and Agriculture Organisation (FAO) (http://www.fao.org/school-food/areas-work/food-environment/en/) and we did not include studies that explored what was available in the curriculum. No studies from this review included nutrition assessments of the food environment that were conducted by a nutritionist or dietitian. We do agree though, that this is an important area of research.
Reviewer 3 Report
A clear conception of school food environment should be provided.
Abstract: should clearly describe how many methods were summaired and what were they, their advantage and disadvantage,ect.
Exclusion criteria: vending machine is a very important way of food provision, especially at schoold, why excluded?
For assessing the quality of methods:
four ceriteria were considered, comprehensiveness, relevance, generalizability and feasibility, how to rate each of it and how to devermine the overall qualtiy?
in addtion to that, their validity and reliabiltiy should be assessed.
Table 3 the information is not consistent and incomplete.
Discussion: according to the title of the manuscript, it should focus the methods per se, however, most of the discussion was about the current situration, improvement of school food evironment, not focus on the methods.
Author Response
Measurement methods used to assess the school food environment: A Systematic Review
To the reviewers and editorial team,
Thank you for your comprehensive review of our manuscript. Thanks to your comments, we feel the manuscript is significantly improved. Please see below your comment, our response and a reference to the manuscript with tracked changes.
With many thanks,
The authors.
Reviewer 3
Comment 1: A clear conception of school food environment should be provided.
Response 1: We have now added our definition of the school food environment.
Reference in the manuscript with tracked changes: Page 2, lines 53-56
The school food environment as defined by the United Nation’s Food and Agriculture Organization (FAO) “refers to all the spaces, infrastructure and conditions inside and around the school premises where food is available, obtained, purchased and/or consumed (for example tuck shops, kiosks, canteens, food vendors, vending machines” [24].
Comment 2: Abstract: should clearly describe how many methods were summarized and what were they, their advantage and disadvantage, etc.
Response 2: Thank you for the suggestion. We have included the number of global studies (n = 38) ( which included 49 methods of measuring the food environment)and what the methods were according to the four dimensions of the food environment (physical environment 47%, policy environment 37% economic 8%, socio cultural 8% environment) which aligns with aim 1 of our study. To further address this concern we have now added more details in the results section details about the number of methods that we assessed.
The number of methods which were assessed was 49, as a number of studies applied more than one methodological approach to measure to school food environment.
In addition, we have added more details about the overall quality, advantages and disadvantages of the methods, as well as other potential methods (added to the discussion, please see reviewer 1’s suggested references).
Please see response to comment 7 below and see page 7 lines 217-250 of the methods.
Reference in the manuscript with tracked changes: please also see Page 6 Lines 214-216 and line 18 of the abstract.
Comment 3: Exclusion criteria: vending machine is a very important way of food provision, especially at school, why excluded?
Response 3: Vending machines in schools do provide a convenient and additional source of food and beverages to children. However, given that a number of countries/cities are implementing healthy vending standards to improve the nutritional quality of foodstuffs available in machines, we felt that this would warrant a separate review of global studies examining the provision of food and beverages in vending machines and would also be quite interesting.
Reference in the manuscript with tracked changes: none.
For assessing the quality of methods:
Comment 4: four criteria were considered, comprehensiveness, relevance, generalizability and feasibility, how to rate each of it and how to determine the overall quality?
Response 4: We have provided Supplementary material Figure S1: Criteria and standards for quality assessment of the school food environment measurement methods which shows how each criterion was applied to each study to give a rating. We have also included how we determined the overall quality (low, medium or high) was applied to each study (Overall assessment in section in the table). We have also referred to the Supplementary material Figure S1 on page 4, line 141.
Reference in the manuscript with tracked changes: please refer to Supplementary material Figure S1.
Comment 5: in addition to that, their validity and reliability should be assessed.
Response 5: We have now included details the validity and reliability for each study in Table 2 and have also include details in the results section.
Manuscript with tracked changes: Page 7 Lines 249-250
Of note, only small number of studies (n = 7) included validity and/or reliability tests for the methods used to measure the school food environment, the details of which are included in Table 2.
Comment 6: Table 3 the information is not consistent and incomplete.
Response 6: We have now added more details regarding the methods for each study, including details about validity and reliability testing to Table 2. We have also extensively reviewed table 2 to ensure it is consistent and complete.
Comment 7: Discussion: according to the title of the manuscript, it should focus the methods per se, however, most of the discussion was about the current situation, improvement of school food environment, not focus on the methods.
Response 7:
Thank you for your comment. We do think it is important to place the reviewed measurement methods in the context of the global public health space and to offer useful suggestions for future practice. However, we have added more details about the methods to the results section (we felt this fitted best here), which includes advantages and disadvantages (as per comment 2).
Manuscript with tracked changes: Page 6 lines 217-255
The most common method used to measure the school food environment was a self-administered questionnaire/survey(n=21) [28,36,39,40,42-45,47,48,51-54,56,58,59,62,64,66,68] to obtain information such as food policy, food purchased or availability from tuckshops or school meals, presence of a tuckshop, if nutrition training was a priority or attitudes to nutrition from either school principals, students, educators or canteen managers. Disadvantages with this method include self-reported bias in favour of desirable rather than actual practice, a low response rate which may not be indicative of true food provision and respondent burden. Also, survey results may not capture realistic practices e.g. food policy as an indicator of the policy environment does not assess the level of effectiveness of a policy [43]. Advantages are the low cost, ease of administration, access to a large number of participants and the reporting of other nutrition practices that may be otherwise overlooked by just reviewing menus.
Six of our included studies utilised observational data collection methods where trained researchers observed food provision in the canteens/tuckshops or school restaurants [37, 38,47,60,65,67]. This data collection method can be highly variable and can subsequently provide inaccurate reflections of food provision and poor generalisability of findings [72]. However, direct observation methods may improve the validity of self-administered surveys and on-site/direct observations are considered the ideal approach in assessing school environmental characteristics [60] and nutrition practices [32].
Canteen menu analysis was utilised by six studies, which involved either obtaining a school menu online or from canteen managers [22,55,61,65,70,71]. Although menu review is an objective measure, it may not be a reliable tool for food provision assessment, as the actual food available may differ from the planned menu and insufficient information may limit the account of portion sizes, types of foods, or pricing [64]. Skilled researchers (e.g. dietitians) in menu coding and analysis are also often required to conduct the research and often menus are only assessed at one point in time, so the certainty of the menus being offered at all times, over the school year is unknown. However, compared to on-site observations, menu reviews are lower in implementation costs and less labour-intensive [64]
Seven studies included semi-structured interviews with school principals or canteen/food service manager [41,46,50,57,60,64,69] and/or focus groups with students [46,64]. This subjective measurement method is expensive due to the labour required to conduct the research. However, rich data can be obtained and other food provision practices may be captured that may be otherwise missed by observational methods.
Of note, only small number of studies (n = 7) included validity and/or reliability tests for the methods used to measure the school food environment, the details of which are included in Table 2.
This manuscript is a resubmission of an earlier submission. The following is a list of the peer review reports and author responses from that submission.
Round 1
Reviewer 1 Report
Comments to Authors
The current study has showed that: 1) there are no standardised methods used to measure the school food environment; 2) robust methods to monitor the school food environment across a range of diverse country contexts is required to provide an understanding of obesogenic school environments.
Authors are kindly requested to emphasize the current concepts about these issues in the context of recent knowledge and the available literature. This articles should be quoted in the References list.
References
A systematic review employing the GeoFERN framework to examine methods, reporting quality and associations between the retail food environment and obesity. Health Place. 2019 May;57:186-199. doi:10.1016/j.healthplace.2019.02.007. Food store environment examination - FoodSee: a new method to study the food store environment using wearable cameras. Glob Health Promot. 2019 Aug 26: 1757975919859575. doi: 10.1177/1757975919859575.